# Population dynamics and species composition of maize field parasitoids attacking aphids in northeastern China

**Ying Zhang, Min-chi Zhao, Jia Cheng, Shuo Liu, Hai-bin Yuan**  *

Department of Plant Protection, Jilin Agricultural University, Changchun, China

* yhb-74@163.com

**Data Availability Statement:** All relevant data are within the manuscript.

**Funding:** This research was funded by the The National Science and Technology major project of

## Abstract

Maize, *Zea mays* L., is the most abundant field crop in China. Aphids are the most economically damaging pest on maize, particularly in the maize agri-ecosystems of Jilin Province, northeastern China. Parasitic wasps are important natural enemies of aphids, but limited information exists about their species composition, richness and seasonal dynamics in northeastern China. In this study, the population dynamics of maize aphids and parasitoid wasps were assessed in relation to each other during the summer seasons of two consecutive years, 2018 and 2019. We selected maize fields in the Changchun, Songyuan, Huinan and Gongzhuling areas of Jilin Province. Four species of aphids were recorded from these maize fields: *Rhopalosiphum padi* (L), *Rhopalosiphum maidis* (Fitch*)*, *Aphis gossypii* Glover and *Macrosiphum miscanthi* (Takahashi). The dominant species in each of the four areas were *R. maids* (Fitch) and *R. padi* in Changchun, *R. padi* in Songyuan, *A. gossypii* and *R. padi* in Huinan, and *A.gossypii* and *R. padi* in Gongzhuling. We delineated a species complex made up of primary parasitoids and hyperparasitoids associated with maize aphids. The primary parasitoids *Lysiphlebus testaceipes*, *Binodoxys communis* and *Aphelinus albipodus* together formed approximately 85.3% of the parasitoid complex. *Pachyneuron aphidis*, *Phaenoglyphis villosa*, *Syrphophagus taeniatus* and *Asaphes suspensus* made up the hyperparasitoids. Of the primary parasitoids, *L. testaceipes* was the dominant species (81.31%). Of the hyperparasitoid group, *P. villosa* was the dominant species (68.42%). Parasitism rates followed the fluctuation of the aphid population. The highest parasitic rate was observed during the peak period of cotton aphids. In this paper, the occurrence dynamics and dominant species of aphids and the dynamics of parasitic natural enemies of aphids in maize fields in Jilin Province are, for the first time, systematically reported. This study provides important information for the establishment and promotion of aphid biological control in maize fields.

## Introduction

Maize is the most extensively cultivated cereal in the world and serves as a staple in many tropical, subtropical and warm temperate countries [1, 2]. Maize, which originated in China, is an important crop in the northern and northeastern regions of the country, by 2050, the predicted

China (Grant No.2019ZX08012004-009), and the National Key Research and Development Program of China (Grant No.2017YFD0300606).The funders had no role in study design, data collection and analysis, decision to publish, or preparation of the manuscript.

**Competing interests:** The authors have declared that no competing interests exist.

demand for maize in the developing world would be doubled [3]. The aphid is one of the most economically damaging pest on maize. The common aphid species in maize fields include *R. padi*, *R. maidis*, *M. miscanthi* and other wheat aphids. Among them, *R. maidis* and *R. padi* have the most economic effects on maize yields. The large number of aphids and the serious damage they cause lead to great losses in maize yield each year [4–6]. In addition to causing direct damage by removing photoassimilates, aphids transmit several destructive maize viruses, including the maize yellow dwarf virus, barley yellow dwarf virus, sugarcane mosaic virus, and cucumber mosaic virus [7]. Damage caused to maize by aphids takes several forms, with resulting yield losses being quite variable from year to year. Growth and yield are reduced through the removal of photosynthates by large numbers of aphids [8]. For example, when the strain rate of maize reaches 25%, direct or indirect losses can reach 12.5% [9], and the most serious losses can be up to 50%, or even no maize production. Short-lived outbreaks have been reported in some areas and have at times caused considerable yield loss, which has led to various control measures [10, 11]. However, control of the aphid-caused damage to maize is challenging due to aphids' high population fecundity rate, the difficulty of applying pesticides, and the ease with which aphids develop resistance to pesticides.

Fortunately, these aphids are attacked by a bundle of natural enemies, including parasitoid wasps, which are an important component of the maize field ecosystem. Biological control services provide an effective strategy for pest suppression and natural enemy promotion. The trophic structure of natural enemies and the organization of food webs in agri-ecosystems have become key issues for biological control [12, 13]. Studies have been conducted on the artificial feeding and release of predatory natural enemies of aphids in the field [14, 15]. However, few have systematically studied the control effect of maize field parasitoids on maize field aphids [16, 17]. Lack of a comprehensive understanding of the parasitoid community in maize fields and of the efficiency of parasitoids in controlling these pests, prevails.

Aphid-primary parasitoid relationships have been extensively documented, but aphids and primary parasitoid populations can also be influenced by hyperparasitoids, of which reports are mostly lacking. For example, on alfalfa crops in Spain, 13 Aphidiinae parasitoid species and 4 species of aphelinids, as well as eight hyperparasitoids were recorded [18]. Cereal aphids are subject to parasitism rates of 30–80% on Danish and New Zealand winter wheat, mostly from *Aphidius ervi* Haliday and *Aphidius rhopalosiphi* (De Stefani-Perez) [19]. In northern China, two primary parasitoid species and ten different hyperparasitoid species were collected from cotton fields [20]. In another study, three primary parasitoid species were found, and the hyperparasitoid guild was more diverse, consisting of 14 species [20, 21]. Two primary parasitoid species and 4 secondary parasitoids were found on corn leaves, consisting of the species *Aphelinus nigritus* (Howard), *A. varipes* (Forester), *Pachyneuron siphonophorae* (Ashmead), *Aphidencyrtus aphidivorus* (Mayr), *Charips* sp., and *Asaphes hicens* (Provancher) [22]. However, research on parasitoids in maize fields is still scarce in northeastern China.

Here, we investigated the parasitoid species composition and the population dynamics of maize aphids and their parasitoids in four maize fields in 2018 and 2019 seasons in northeastern China. This work will support future research on the species characterization and community composition of maize aphid parasitoids in China.

## Materials and methods

### Location

The experimental fields were maize fields located at the Changchun(125.2963˚E, 43.9646˚N), Songyuan,(124.8315˚E, 45.1474˚N), Huinan(126.2683˚E, 42.6363˚N) and Gongzhuling (124.8294˚E, 43.5108˚N) research station in Jilin Province, China (Fig 1). The fields were

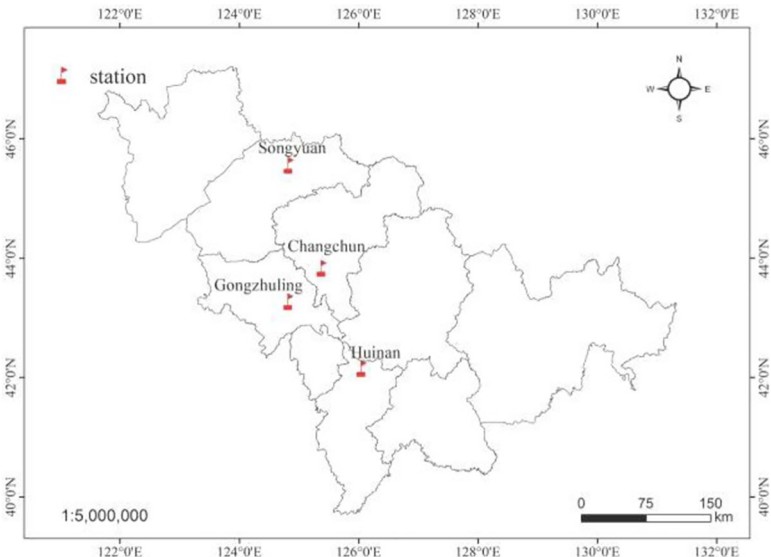

**Fig 1. The map of experimental fields in four maize fields of Jilin province, northeastern China.**

cultivated according to the management practices of local farmers, with no crop rotation. No pesticides were applied during the growing season. These fields were all cooperative plots between the local farmers and Jilin Agricultural University. There were no specific permissions activities were required and did not involve endangered or protected species in these experimental fields.

## Field survey

In open field survey, sampling was started in early July in 2018 and 2019 when winged aphids migrated on maize plants, at ten days interval throughout the all seasons until maize harvest. A five-point sampling method was used to set up three groups of sample repeats in each area, each of which was repeated at five points, and 20 maize plants were randomly selected at each point. Sampling during the open field survey was carried out as follow: Each plant (as a whole) was visually examined and counted for the aphids and aphid parasitoids. Mummy samples were collected randomly in 2019 from various open field plots (when parasitoid densities were at high levels) for further identification of parasitoids. The collected mummies were brought back to the laboratory and placed individually in petrie dish in a climatic chamber (25˚C, 65% RH and 16:8h/ L:D). The emerged parasitoids were identified [16].

## Data analysis

Parasitism rates were calculated by dividing the number of aphid mummies by the sum of mummified and living aphids. The field data obtained were preliminarily sorted in Excel and analyzed by SPSS 17.0.

## Aphid advantage analysis method

Data were analyzed using Berger Parker's dominance [23] and Simpson's dominant concentration [24] methods, using this formula,

$$B = n_{max}/N, \ C = \sum(n_i/N)^2$$

Where B is aphid population dominance, C is the aphid community dominance concentration

index, $n_{max}$ is the number of aphid species in the aphid community, N is the total number of organisms in the community, and $n_i$ is the number of individuals of the first species. The greater the population dominance is the greater the individual difference between species in the community, the prominent the dominant species, the fiercer the competition among species, and the greater unstable state of the community. The dominant concentration index is the concentration trend of the number of individuals in the reaction species.

## Results

### Identification of aphid species in maize fields in Jilin province

Four different aphids species were identified under a microscope: *R. padi* (Fig 2A), *R. maids* (Fig 2B), *A. gossypii* (Fig 2C), and *M. miscanthi* (Fig 2D). For a long time in China, the aphid *M. miscanthi* had been misidentified as *Macrosiphum avenae* (Fabricius). However, Guangxue (1999) showed that the distribution of *M. avenae* was only limited to some areas in Yili, Xinjiang, China. Thus *M. miscanthi* is the correct identification for those identified as *M. avenae* in China in the past [25].

### Dominant aphid species and population dynamics of maize fields in Changchun

Different aphid species caused damage to maize at different times in the Changchun area. The dominant aphid species were also different in different periods. In early July 2018–2019, the presence of *R. padi*, *A. gossypii* and *M. miscanthi* were recorded, of which the dominant species was *R. padi*. The highest population dominance, B, was 0.87, and the advantage concentration index, C, was 0.77. However, the presence of *R. maids* was recorded in early August, and by the end of August to early September, its population had become the largest. Thus, the

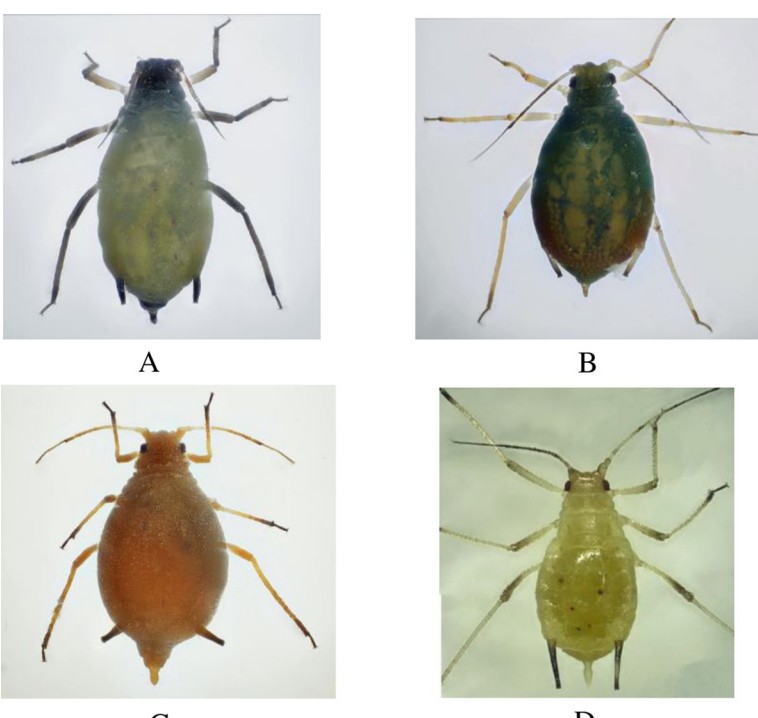

**Fig 2. Morphology of 4 maize field aphids under ultraclear microscopy.** (A) *R. padi*; (B) *R. maids*; (C) *A. gossypii*; (D) *M. miscanthi*.

**Table 1. The population dynamics and dominance of aphids in Changchun area maize.**

|  | Date | *R.maids* | Percentage(%) | *R. padi* | Percentage(%) | *A. gossypii* | Percentage(%) | *M. miscanthi* | Percentage(%) | B | C |
|---|---|---|---|---|---|---|---|---|---|---|---|
| **2018** | Early Jul | 0.00 | 0.00% | 333.33 | 63.69% | 190.00 | 36.31% | 0.00 | 0.00% | 0.64 | 0.54 |
|  | Mid-Jul | 0.00 | 0.00% | 334.67 | 46.55% | 384.33 | 53.45% | 0.00 | 0.00% | 0.53 | 0.50 |
|  | Late Jul | 0.00 | 0.00% | 369.33 | 65.91% | 188.00 | 33.55% | 3.00 | 0.54% | 0.66 | 0.55 |
|  | Early Aug | 5.00 | 0.70% | 544.00 | 75.66% | 155.00 | 21.56% | 15.00 | 2.09% | 0.76 | 0.62 |
|  | Mid-Aug | 1854.00 | 69.08% | 783.00 | 29.17% | 47.00 | 1.75% | 0.00 | 0.00% | 0.69 | 0.56 |
|  | Late Aug | 4344.00 | 83.18% | 878.50 | 16.82% | 0.00 | 0.00% | 0.00 | 0.00% | 0.83 | 0.72 |
|  | Early Sept | 6153.57 | 89.21% | 744.00 | 10.79% | 0.00 | 0.00% | 0.00 | 0.00% | 0.89 | 0.81 |
|  | Mid-Sept | 5250.00 | 90.98% | 520.50 | 9.02% | 0.00 | 0.00% | 0.00 | 0.00% | 0.91 | 0.84 |
| **2019** | Early Jul | 0.00 | 0.00% | 187.97 | 60.84% | 121.00 | 39.16% | 0.00 | 0.00% | 0.61 | 0.52 |
|  | Mid-Jul | 0.00 | 0.00% | 321.35 | 57.81% | 234.56 | 42.19% | 0.00 | 0.00% | 0.58 | 0.51 |
|  | Late Jul | 0.00 | 0.00% | 393.33 | 86.70% | 55.00 | 12.12% | 5.33 | 1.74% | 0.87 | 0.77 |
|  | Early Aug | 66.67 | 14.70% | 114.00 | 52.94% | 34.67 | 7.64% | 0.00 | 0.00% | 0.53 | 0.31 |
|  | Mid-Aug | 235.00 | 46.35% | 249.00 | 49.11% | 23.00 | 4.54% | 0.00 | 0.00% | 0.49 | 0.46 |
|  | Late Aug | 808.33 | 84.17% | 152.00 | 15.83% | 0.00 | 0.00% | 0.00 | 0.00% | 0.84 | 0.73 |
|  | Early Sept | 566.33 | 86.29% | 90.00 | 13.71% | 0.00 | 0.00% | 0.00 | 0.00% | 0.86 | 0.76 |
|  | Mid-Sept | 90.00 | 72.00% | 35.00 | 28.00% | 0.00 | 0.00% | 0.00 | 0.00% | 0.72 | 0.60 |

dominant aphid species in the maize field gradually changed from *R. padi* to *R. maids*. The population dominance B gradually rose from 0.49 to 0.91, and the advantage concentration index C increased from 0.46 to 0.84. (Table 1)

## Dominance and population dynamics of aphid species in maize fields in Songyuan

The presence of *R. padi*, *A. gossypii* Glover and *M. miscanthi* were recorded in early July, 2018 and 2019. The presence of *R. maids* was recorded in early August. From July to September, the dominant species was *R. padi*. The highest population dominance B was 0.99, and the advantage concentration index C was 1. In mid-August and late August, the population of *R. padi* reached a maximum. The aphid density per 100 plants was greater than 30,000 in 2018, and in 2019, the number of aphids per 100 plants reached more than 15,000. In the whole maize growth period, a small number of *R. maids* and *M.miscanthi* were recorded. (Table 2)

## Dominances and population dynamics of aphid species in maize fields in Huinan

The presence of *R. padi* and *A. gossypii* were recorded in early July 2018 and 2019 and that of *R.maids* (Filch) was recorded in late August. In July, the dominant species was *A. gossypii*. The highest population dominance B was 0.77, and the advantage concentration index C was 0.65. *R. padi* became the dominant species in early August to mid-September. The highest population dominance B was 0.95 and the advantage concentration index C was 0.91. The population of *R. padi* reached a maximum in early September. The aphid density per 100 plants was 9326.25 in 2018. In 2019, the number of aphids per 100 plants reached 4538.89. In the whole maize growth period, a small number of *M. miscanthi* were recorded (Table 3).

## Dominance and population dynamics of aphid species in maize fields in Gongzhuling

The presence of *R. padi*, *A. gossypii* and *M. miscanthi* were recorded in the first ten days of July 2018 and 2019. The presence of *R. maids* was recorded in August. *A. gossypii* became the

**Table 2. The population dynamics and dominance of aphids in Songyuan area maize field in 2018 and 2019.**

| | Date | R. maids | Percentage(%) | R. padi | Percentage(%) | A. gossypii | Percentage(%) | M. miscanthi | Percentage(%) | B | C |
|---|---|---|---|---|---|---|---|---|---|---|---|
| 2018 | Early Jul | 0.00 | 0.00% | 31.67 | 72.52% | 12.00 | 27.48% | 0.00 | 0.00% | 0.73 | 0.60 |
| | Mid-Jul | 0.00 | 0.00% | 45.89 | 43.32% | 34.60 | 32.66% | 25.45 | 24.02% | 0.43 | 0.35 |
| | Late Jul | 0.00 | 0.00% | 63.00 | 65.78% | 0.00 | 0.00% | 32.78 | 34.22% | 0.66 | 0.55 |
| | Early Aug | 36.00 | 6.59% | 502.00 | 91.86% | 0.00 | 0.00% | 8.50 | 1.56% | 0.92 | 0.85 |
| | Mid-Aug | 96.00 | 3.08% | 3005.83 | 96.53% | 12.00 | 0.39% | 0.00 | 0.00% | 0.97 | 0.93 |
| | Late Aug | 41.00 | 0.11% | 38857.10 | 99.89% | 0.00 | 0.00% | 0.00 | 0.00% | 0.99 | 1.00 |
| | Early Sept | 165.00 | 45.21% | 175.00 | 47.95% | 25.00 | 6.85% | 0.00 | 0.00% | 0.48 | 0.44 |
| | Mid-Sept | 75.00 | 88.24% | 10.00 | 11.76% | 0.00 | 0.00% | 0.00 | 0.00% | 0.88 | 0.79 |
| 2019 | Early Jul | 0.00 | 0.00% | 61.67 | 69.42% | 26.67 | 30.02% | 0.50 | 0.56% | 0.69 | 0.57 |
| | Mid-Jul | 0.00 | 0.00% | 89.78 | 49.48% | 57.89 | 31.90% | 33.78 | 18.62% | 0.49 | 0.38 |
| | Late Jul | 0.00 | 0.00% | 170.00 | 81.77% | 0.00 | 0.00% | 37.89 | 18.23% | 0.82 | 0.70 |
| | Early Aug | 56.67 | 14.40% | 308.33 | 78.34% | 15.00 | 3.81% | 13.56 | 3.45% | 0.78 | 0.64 |
| | Mid-Aug | 43.00 | 0.27% | 16166.00 | 99.73% | 0.00 | 0.00% | 0.00 | 0.00% | 0.99 | 0.99 |
| | Late Aug | 0.00 | 0.00% | 270.00 | 94.74% | 15.00 | 5.26% | 0.00 | 0.00% | 0.95 | 0.90 |
| | Early Sept | 0.00 | 0.00% | 96.83 | 66.86% | 48.00 | 33.14% | 0.00 | 0.00% | 0.67 | 0.56 |
| | Mid-Sept | 0.00 | 0.00% | 74.00 | 18.55% | 325.00 | 81.45% | 0.00 | 0.00% | 0.81 | 0.70 |

dominant species in July-August. Its population dominance B was 1, and the advantage concentration index C was 1. The dominant aphid species in the maize field gradually changed from *A. gossypii* to *R.padi* from August to September, and the population dominance B of the latter rose from 0.70 to 0.99. The advantage concentration index C rose from 0.57 to 0.99 while the population increased. The aphids *R. maids* and *M. miscanthi* were present in small numbers (Table 4).

## Species composition of parasitic wasps

In 2019, a total of 1,448 parasitoid specimens were collected including three primary parasitoid species: *B. communis*, *L. testaceipes* and *A. albipodus*; and four different hyperparasitoid

**Table 3. The population dynamics and dominance of aphids in the Huinan area maize field in 2018 and 2019.**

| | Date | R.maids | Percentage(%) | R.padi | Percentage(%) | A. gossypii | Percentage(%) | M. miscanthi | Percentage(%) | B | C |
|---|---|---|---|---|---|---|---|---|---|---|---|
| 2018 | Early Jul | 0.00 | 0.00% | 48.89 | 28.83% | 120.67 | 71.17% | 0.00 | 0.00% | 0.71 | 0.59 |
| | Mid-Jul | 0.00 | 0.00% | 101.25 | 39.27% | 156.34 | 60.63% | 0.25 | 0.10% | 0.61 | 0.52 |
| | Late Jul | 0.00 | 0.00% | 68.00 | 54.54% | 56.67 | 45.46% | 0.00 | 0.00% | 0.55 | 0.50 |
| | Early Aug | 0.00 | 0.00% | 72.00 | 74.40% | 24.78 | 25.60% | 0.00 | 0.00% | 0.74 | 0.62 |
| | Mid-Aug | 0.00 | 0.00% | 325.65 | 89.88% | 36.68 | 10.12% | 0.00 | 0.00% | 0.90 | 0.82 |
| | Late Aug | 222 | 29.30% | 507.86 | 67.02% | 26.78 | 3.53% | 1.11 | 0.15% | 0.67 | 0.45 |
| | Early Sept | 2451 | 20.69% | 9326.25 | 78.73% | 68.78 | 0.58% | 0.00 | 0.00% | 0.79 | 0.62 |
| | Mid-Sept | 913.33 | 33.51% | 1809.50 | 66.39% | 2.87 | 0.11% | 0.00 | 0.00% | 0.66 | 0.44 |
| 2019 | Early Jul | 0.00 | 0.00% | 78.75 | 22.52% | 270.67 | 77.39% | 0.33 | 0.10% | 0.77 | 0.65 |
| | Mid-Jul | 0.00 | 0.00% | 211.83 | 41.74% | 295.67 | 58.26% | 0.00 | 0.00% | 0.58 | 0.51 |
| | Late Jul | 0.00 | 0.00% | 268.63 | 36.11% | 475.25 | 63.89% | 0.00 | 0.00% | 0.64 | 0.54 |
| | Early Aug | 0.00 | 0.00% | 916.44 | 73.93% | 323.22 | 26.07% | 0.00 | 0.00% | 0.74 | 0.61 |
| | Mid-Aug | 0.00 | 0.00% | 1467.25 | 83.60% | 286.45 | 16.32% | 1.33 | 0.08% | 0.84 | 0.73 |
| | Late Aug | 14.38 | 0.7% | 1922.88 | 94.25% | 100.63 | 4.93% | 2.38 | 0.12% | 0.94 | 0.89 |
| | Early Sept | 108.78 | 2.28% | 4538.89 | 95.13% | 123.67 | 2.59% | 0.00 | 0.00% | 0.95 | 0.91 |
| | Mid-Sept | 469.33 | 27.14% | 1236.67 | 71.51% | 23.46 | 1.36% | 0.00 | 0.00% | 0.72 | 0.51 |

**Table 4. The population dynamics and dominance of aphids in Gongzhuling area maize field in 2018 and 2019.**

|  | Date | *R. maids* | Percentage(%) | *R. padi* | Percentage(%) | *A. gossypii* | Percentage(%) | *M. miscanthi* | Percentage(%) | B | C |
|---|---|---|---|---|---|---|---|---|---|---|---|
| 2018 | Early Jul | 0 | 0% | 8.33 | 2.19% | 371.67 | 97.76% | 0.17 | 0.04% | 0.98 | 0.96 |
|  | Mid-Jul | 0 | 0% | 83.33 | 9.69% | 768.56 | 89.39% | 7.89 | 1% | 0.89 | 0.81 |
|  | Late Jul | 0 | 0% | 439.89 | 38.58% | 666.44 | 58.45% | 33.78 | 2.96% | 0.58 | 0.49 |
|  | Early Aug | 35 | 2.32% | 647.11 | 42.94% | 847.88 | 56.26% | 12 | 0.79% | 0.56 | 0.50 |
|  | Mid-Aug | 47.5 | 1.730% | 678.55 | 24.70% | 2067.33 | 75.26% | 1.09 | 0% | 0.75 | 0.63 |
|  | Late Aug | 25.37 | 0.92% | 1316 | 69.81% | 543.67 | 29% | 0 | 0% | 0.70 | 0.57 |
|  | Early Sept | 31.5 | 1.46% | 1897.5 | 87.68% | 235 | 11% | 0 | 0% | 0.88 | 0.78 |
|  | Mid-Sept | 5.97 | 0.27% | 2104.34 | 93.89% | 131 | 6% | 0 | 0% | 0.94 | 0.88 |
| 2019 | Early Jul | 0 | 0% | 0 | 0.00% | 22.67 | 100.00% | 0 | 0% | 1 | 1.00 |
|  | Mid-Jul | 0 | 0.00% | 6.33 | 1.89% | 264 | 78.73% | 65 | 19.38% | 0.79 | 0.66 |
|  | Late Jul | 0 | 0.00% | 0 | 0.00% | 461.33 | 93.89% | 30 | 6.11% | 0.94 | 0.89 |
|  | Early Aug | 15 | 0.45% | 591.67 | 17.91% | 2696.67 | 81.63% | 0 | 0.00% | 0.82 | 0.70 |
|  | Mid-Aug | 5 | 0.07% | 612.33 | 8.04% | 7002.33 | 91.90% | 0 | 0.00% | 0.92 | 0.85 |
|  | Late Aug | 0 | 0.00% | 1224 | 33.50% | 2430 | 66.50% | 0 | 0.00% | 0.67 | 0.55 |
|  | Early Sept | 27 | 1.16% | 2235 | 96.29% | 56 | 2.41% | 3 | 0.13% | 0.96 | 0.93 |
|  | Mid-Sept | 0 | 0.00% | 2530 | 99.49% | 13 | 0.51% | 0 | 0.00% | 0.99 | 0.99 |

species: *P. aphidis*, *P. villosa*, *S. taeniatus* and *A. suspensus* (Fig 3). They passed molecular and morphological identifications. The proportion of primary parasitoids was 85.3%, and the proportion of hyperparasitoid species was 14.7%. *L. testaceipes* was the dominant primary parasitoid formed (81.31%) while *P.villosa* was the dominant hyperparasitoid (68.42%) (Fig 4). The parasitic rate increased with increase in aphid occurrence and vice versa. At the peak of aphid occurrence in mid-August, a higher parasitic rate of 1.79% was recorded for the parasitoids. Thereafter, the parasitic rate decreased to 0 by mid-September (Fig 5).

## Discussion

Since the 1950s, aphids have been considered as the main pests affecting maize in many countries and therefore systematic and comprehensive research on their biological characteristics, the relationship between their reproduction and their hosts, their species composition, and the influence of the environment on their reproduction have been carried out [9, 26]. However, compared to that in other countries, research on aphids in China started later. In recent years, aphids have caused serious damage to maize in some fields in China, which seriously endangers maize production. The study of aphids' occurrence and the cause of their outbreaks have attracted the attention of many agricultural researchers [27]. We investigated the occurrence dynamics and species composition of aphids and parasitoids in four maize-growing areas in Jilin Province over two years. This study revealed that 4 species of aphids occurred in the maize fields in this province: *R. padi*, *R. maidis*, *A. gossypii* and *M. miscanthi*. *A. gossypii* was found on the back of the lower leaves of maize plants, and *R. maidis* occurred first in the lower part of the maize and the lower leaf, and the gradually transferred upward to the female ear of maize when it aged. *R. padi* was found in the male and female ear bract leaves, and *M. miscanthi* on the lower stem of the maize plant. This result is consistent with the results of other research conducted in China. Yang et al. (2013) reported that the species of maize aphids found in Jilin Province were *R. maids* and *R. padi* [28]. In Haerbin province, the aphid species reported as found in maize fields were *R. maids*, *R. padi*, *A. gossypii*, *M. miscanthi* and *Schizaphis graminum* (Rondani) [27, 29, 30]. However, these reports differ with the results from studies conducted in maize fields in other countries. For instance, *R. padi* (L.), *Sitobion avenae*

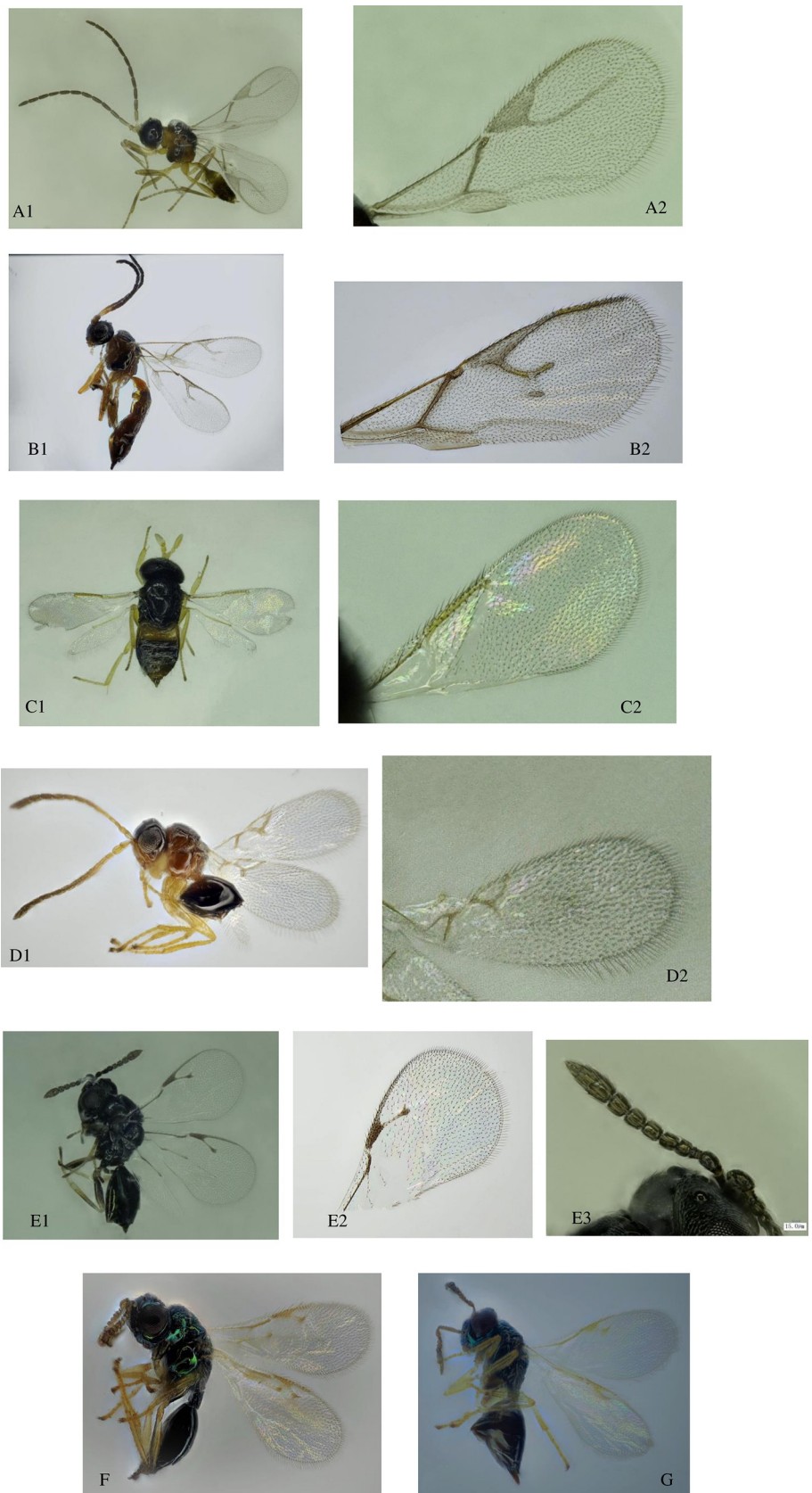

**Fig 3. Morphology of 7 parasitic wasps under an ultraclear microscope.** (A1, A2) *B. communis*; (B1, B2) *L. testaceipes*; (C1, C2) *A. albipodus*; (D1, D2) *P. aphidis*; (E1, E2, E3) *P. villosa*; (F) *S. taeniatus*; (G) *A. suspensus*.

(F.) and *Metopolophium dirhodum* (Walker) were the most abundant aphid species attacking maize in the northeastern Iberian Peninsula [31]. Thirteen species, including *Aphis craccivora Koch*, *Aphis fabae Scopoli*, *A. gossypii*, *Aphis maidiradicis Forbes*, *Hysteroneura setariae* (Thomas), *Macrosiphum euphorbiae* (Thomas), *Metopolophium dirhodium* (Walker), *Myzus persicae* (Sulzer), *R.maidis*, *R.padi*, *Sipha flava* (Forbes), *Schizaphis graminum* (Rondani), and *Sitobion avenae* (Fabricius) were associated with maize in the United States [32]. *R.maidis* and *R.padi* were reported as the two predominant aphid species on maize in the United States, Iowa [33, 34]. However, we found that the dominant aphid species was different in each of the four maize fields. The dominant species in the Gongzhuling maize field was the cotton aphid, which was the first time that this species has been reported in this area. The difference in aphid species occurrence may be related to the regional differences in temperature and surrounding crops.

The distribution of parasitic wasps is almost nationwide, distributed in northern China, namely in the Jilin, Hebei, Shanxi and Henan Provinces [35]. Parasitoids have been broadly studied in China [36]. However, there is little research and few reports on parasitoids in maize fields. Archer (1972) only collected parasitic wasps from *R. maids* but did not conduct any research on maize fields. Our study reports on the parasitiods found in maize fields: of the four

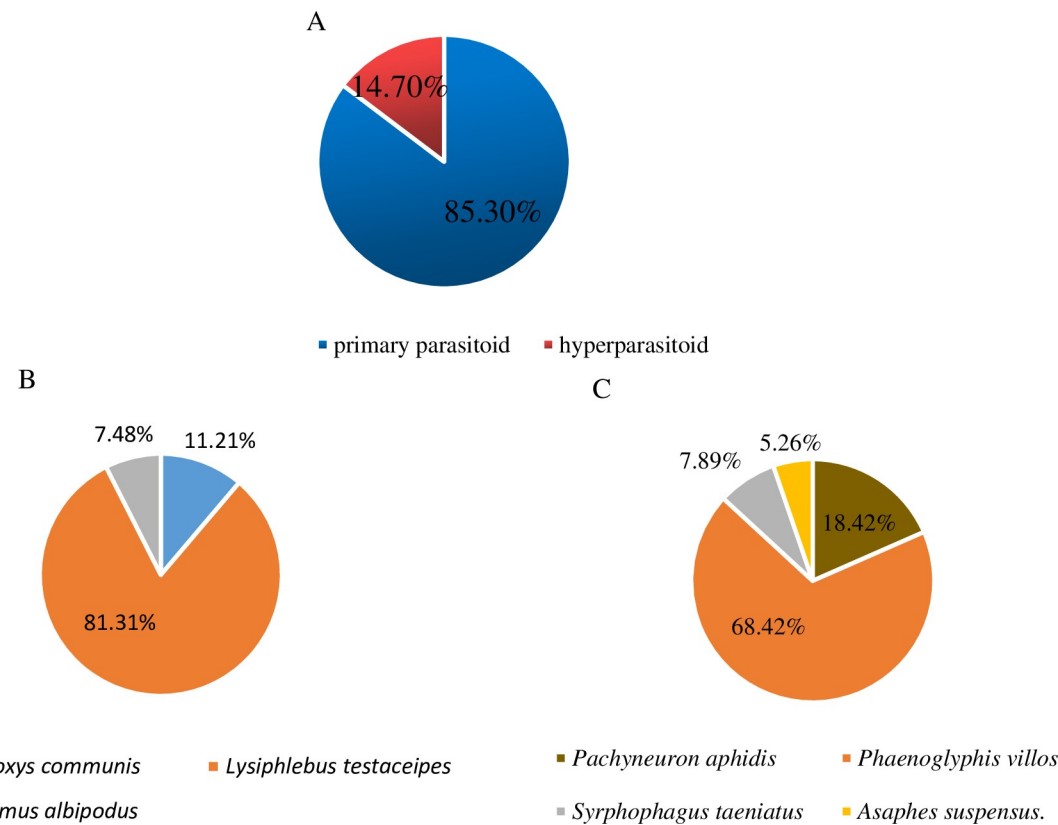

**Fig 4. Composition of the parasitoids associated with aphids in maize fields in Jilin Province in 2019.** (A) Composition of the parasitoids. (B) Composition of the primary parasitoids. (C) Composition of the hyperparasitoids.

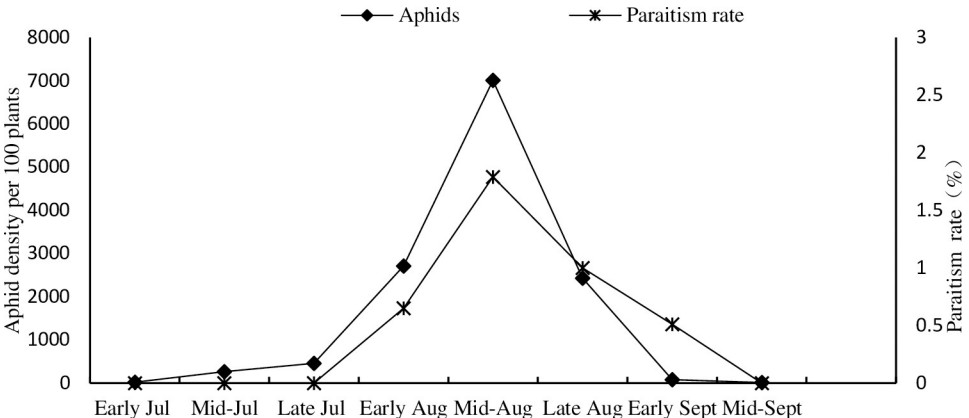

**Fig 5. Maize field aphids density and parasitism dynamics.**

maize fields chosen, most parasitoids occurred in the Gongzhuling area, while fewer parasitoids occurred in other areas. Analysis of the parasitoid species composition was therefore limited to Gongzhuling. The three primary parasitoid species included *L. testaceipes*, *B. communis*, and *A. albipodus*. There were four different hyperparasitoid species, *P. aphidis*, *P. villosa*, *S. taeniatus* and *Asaphes suspensus*. Most of these parasitoids were found on *A. gossypii* and *M.miscanthi*. In many respects, the *L. testaceipes* establishment in southern France and other areas of the Mediterranean region may be considered a useful model for biological control and integrated pest management [37]. The results showed that the aphid species, *R. maidis* could be used as an alternative host for the reproduction of the parasitoid *L. testaceipes* on sorghum [38]. In northern China, three primary parasitoid species and 12 hyperparasitoids on *A. gossypii* were recorded in cotton fields [20], while three parasitoids and 15 hyperparasitoids were collected in wheat fields [37].

Maize aphid population densities were low during this study earlier on and peaked midway through the maize growing period. Parasitism rates increased throughout nearly the entire maize growing season and followed the trends in growth and decline of the aphid populations, a typical seasonal dynamic of aphid-parasitoid interactions [39]. However, the highest parasitic rate was only 1.79% in the whole maize growth cycle, which indicated that the field parasitoids may not effectively control the occurrence of aphids.

## Conclusions

Our results showed that aphid species in maize fields of Jilin Province, northeastern China included *R. padi*, *R. maidis*, *A. gossypii* and *M. miscanthi*. Also, species dominance and occurrence dynamics of aphids in maize fields were different between the different maize fields. Three primary parasitic wasps and five secondary parasitic wasps were recorded in addition. The primary parasitoids (85.3%) included *L. testaceipes*, *B. communis* and *A. albipodus*. The hyperparasitoids (14.7%) were *P. aphidis*, *P. villosa*, *S. taeniatus* and *A. suspensus*. *L. testaceipes* was the dominant primary parasitoid species (81.31%) while *P.villosa* was the dominant hyperparasitoid species (68.42%). The parasitic rate increased with increasing aphid occurrence and vice versa. The highest parasitic rate (1.79%) was recorded at the peak of aphid occurrence in mid-August. Our next focus is to conduct laboratory studies on the parasitic function to clarify the reasons for the low parasitic rate recorded in the field and to continue to improve understanding of the parasitic bee species in the maize field.

## Acknowledgments

We are much thankful to our partners for their support in sample collection, insect rearing and laboratory assays. We also appreciate the College of Plant Protection, Jilin Agricultural University, providing test facilities.

## Author Contributions

**Conceptualization:** Ying Zhang, Hai-bin Yuan.

**Data curation:** Shuo Liu.

**Investigation:** Ying Zhang, Min-chi Zhao, Jia Cheng.

**Writing – original draft:** Ying Zhang.

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
