## [Decision Letter · Decision Letter 0]

18 Nov 2020

PONE-D-20-32389

Species composition and dynamics of aphids and parasitoid wasps in maize fields in Northeastern China

PLOS ONE

Dear Dr. yuan,

Thank you for submitting your manuscript to PLOS ONE. After careful consideration, we feel that it has merit but does not fully meet PLOS ONE’s publication criteria as it currently stands. Therefore, we invite you to submit a revised version of the manuscript that addresses the points raised during the review process.

We look forward to receiving your revised manuscript.

Kind regards,

Yulin Gao

Academic Editor

PLOS ONE

Review Comments to the Author

Reviewer #1: The authors calculated the maize aphids population dynamics in the summer of 2018 and 2019, and the population dynamics of maize aphids and parasitoid wasps were assessed in relation to each other during the summer seasons in 2019.In this study, the population dynamics of maize aphids and parasitoid wasps were systematically reported for the first time in Jilin, which provided important information for the biological control of maize aphids. However, some revisions are needed before the article will be considered for publication.

Comments:

1. The authors should carefully check the References sections, ensure that the formatting of author names and scientific species names were standardized, for example references 15, 31, 32. Moreover, some reference cited is not reporting on M. miscanthi, reference 20.

2. Page 10/18: What molecular method does the authors use to distinguish the different species? The DNA-based taxonomic methods can clarify species relationships, sequence variation in a partial segment of the mitochondrial COI gene is highly effective for identifying species, so if the authors using mitochondrial COI as a DNA barcode to distinguish different species, I suggest that the authors list COI gene sequences of the species mentioned in the manuscript.

3. Please unify the font format in Table 3.

4. In Abstract section, the population dynamics of maize aphids and parasitoid wasps were calculated in two consecutive years, 2018 and 2019, but in Results section, I only find the data for parasitoid wasps in 2019 years, please explain the reason.

5. Why did the authors assess the relationship between aphids and parasitoid wasps, what about other parasitic predators of aphids?

Reviewer #2: Reviewer comments for MS ID: PONE-D-20-32389

May revise title as “Population dynamics and species composition of maize field parasitoids attacking aphids in Northeastern China”

Introduction

Staple food

…….regions of the country. By 2050, the predicted demand for maize in the developing world would be doubled.

Can add any reference in world or china for losses caused by aphids in Maize? For this “The large number of aphids and the serious damage they cause lead to great losses in maize yield each year.”

Rephrase this “Damage caused to maize by aphids takes

several forms, with resulting yield losses being quite variable from year to year.

Growth and yield are reduced through the removal of photosynthates by large

numbers of aphids[5].”

higher population fecundity rate,

suite can be replaced with “bundle of”

may add the references for this “However, few have systematically studied the control effect of maize field parasitoids on maize field aphids.”

May add reference Ali et al., 2016 (Sci Rep), 2018 (AGEE) for “Aphid-primary parasitoid relationships have been extensively documented,”

Ali, Abid, Nicolas Desneux, Yanhui Lu, and Kongming Wu, 2018. Key aphid natural enemies showing positive effects on wheat yield through biocontrol services in northern China. Agriculture, Ecosystems & Environment. 266: 1-9. Impact Factor: 3.541. DOI: https://doi.org/10.1016/j.agee.2018.07.012

Abid Ali, Nicolas Desneux, Yanhui Lu, Bing Liu, Kongming Wu, 2016. Characterization of the natural enemy community attacking cotton aphid in the Bt cotton ecosystem in northern China. Scientific Reports. Impact Factor: 4.122. doi:10.1038/srep24273

Remove space between aphelinids and , “and four species of aphelinids,”

Revise as “ten different hyperparasitoid species were collected from cotton fields[15].”

Keep the same trend for all scientific names as for this or otherwise

Aphidius ervi Haliday and Aphidius rhopalosiphi (De Stefani-Perez) [14]

Two primary parasitoid species and 4 secondary parasitoids were found on corn leaves, consisting of the species Aphelinus nigritus (Howard), A. varipes (Forester), Pachyneuron

siphonophorae (Ashmead), Aphidencyrtus aphidivorus (Mayr), Charips sp., and

Asaphes hicens (Provancher)[17].

However, research on parasitoids in maize fields is still scarce. You might mean in northeastern? Or also one factor could be the change in transgenic maize introduction and now different species composition or dynamics or may be due to less area under cotton in northern China?

Revise as “and their parasitoids in three maize fields during 2018 and 2019 seasons in northeastern China.”

Meterials and methods

Location

Here authors mentioned fours locations while there were 3 mentioned in objevtives so cross match. The experimental fields were maize fields located at the Changchun(E125.35 °,N

43.88°), Songyuan,(E124°48′28", N45°10′09"), Huinan(E125°58′49", N42°16′19")

and Gongzhuling(E125°58′49", N42°16′19") research station in Jilin Province, China

Could you add a map with these locations showing the northeastern locations

Field Survey

You may follow the paper published by Ali et al 2016/2018 for experimental design or sampling in details. You shall add the five sampling method reference as (Ali et al., 2018, 2016).

Tables could be represented well in graphs rather tables and also tables could work for percentage.

There is contradiction in objectives, methodology and results sections so it should be fully addressed in revised versions

Fig3. Can follow the published work by Yang Fang et al, on wheat-parasitoids

Conclusions

Is it Jilin province or norther eastern?

And give future suggestions in the conclusion section

Acknowledgements

Revise as “We are much thankful to our partners for their support in sample collection, insect rearing and laboratory assays. This research was funded”

References

Please cross match references and check carefully according to the format and spelling mistakes like this “15. Yang F, Wu YK, Xu L, Wang Q, Yao ZW, ?iki? V, et al. Species composition and

richness of aphid parasitoid wasps in cotton fields in northern China. entific Reports. 2017;7(1):97-99.”

---

## [Author Response · Author response to Decision Letter 0]

24 Nov 2020

Dear Editors,

Thanks for your decision, and all of us are grateful for valuable comments from two reviewers. We especially appreciate their responsible spirit, and we will try our best to revise our article according to reviewers and resubmit our paper again. We have made changes in accordance with the PLOS ONE's style requirements of the journal. We thoroughly copyedit our manuscript for language usage, spelling, and grammar. 

Data Availability Statement: We have no supporting information files. All relevant data are within the manuscript. 

The answers to reviewers’ questions are followed:

Reviewer #1: Thanks for your valuable comments. 

1. For your question 1 “The authors should carefully check the References sections, ensure that the formatting of author names and scientific species names were standardized, for example references 15, 31, 32. Moreover, some reference cited is not reporting on M. miscanthi, reference 20.”

We carefully checked the references in the revised manuscript, ensure that the formatting of author names and scientific species names were standardized

2. For your question 2. “What molecular method does the authors use to distinguish the different species.” 

We mainly used morphological identification method to identify aphids and their parasitic wasps. One of them could not be identified through morphology, so we identified it combining molecular identification. We use mitochondrial COI as a DNA barcode. Lep F1（5’-3’ATT CAA CCA ATC ATA AAG ATA TTG G）and Lep R1（5’-3’TAA ACT TCT GGA TGT CCA AAA AAT CA）. After the gene sequence comparison, we found that Lysiphlebus testaceipes with a gene sequence similarity of 92.55% consisted with the result of morphological identification.

3. For your question 3. “Please unify the font format in Table 3.”

We carefully unified the font format in Table 3 in the revised manuscript.

4. For your question 4. “In Abstract section, the population dynamics of maize aphids and parasitoid wasps were calculated in two consecutive years, 2018 and 2019, but in Results section, I only find the data for parasitoid wasps in 2019 years, please explain the reason.”

We have already conducted the population dynamic analysis of aphids and parasitic wasps in 2018-2019, but in 2018, the number of parasitic wasps was very few because of weather and sample site selection issues.

5. For your question 5. “Why did the authors assess the relationship between aphids and parasitoid wasps, what about other parasitic predators of aphids?”

Because parasitic wasps were limited information exists about their species composition, richness and seasonal dynamics in northern China. This study provides important information for the establishment and promotion of aphid biological control in maize fields. During our two-year research, we also conducted a population dynamic survey of predatory natural enemies. Our article mainly studies the relationship between parasitic natural enemies and aphids in maize fields. Nextly, we will focus on the relationship between predatory natural enemies and aphids based on your suggestions.

Reviewer #2: Thanks for your valuable comments.

1. In accordance with your suggestion, the title revised as “Population dynamics and species composition of maize field parasitoids attacking aphids in Northeastern China.”

2. We revised the sentence “and it is predicted that, by 2050, the demand for maize in the developing world would will double” to “by 2050, the predicted demand for maize in the developing world would be doubled.”

3. “Can add any reference in world or china for losses caused by aphids in Maize?” 

Some references have been added based on your suggestions. (Huang W, L K. Comprehensive control technology of Rhopalosiphum maidis. Bulletin of Agricultural Science and Technology. 1997;7:31-33. Wang Y, Xu J, Su L. Spatial dynamics of Rhopalosiphum maidis population. Journal of Northwest A & F University. 2002;4:55-58. Patil V. P. Population dynamics, yield losses and chemical control of aphid Rhopalosiphum maidis(Fitch) on maize. College of Agriculture, Junagadh. 2012;7:107.)

4. We revised “higher population fecundity” to “higher population fecundity rate”.

5. We replaced “suite” to “bundle of”.

6. We added the references for this “However, few have systematically studied the control effect of maize field parasitoids on maize field aphids.”(Ali, Abid, Nicolas Desneux, Yanhui Lu, and Kongming Wu, 2018. Key aphid natural enemies showing positive effects on wheat yield through biocontrol services in northern China. Agriculture, Ecosystems & Environment. 266: 1-9. Abid Ali, Nicolas Desneux, Yanhui Lu, Bing Liu, Kongming Wu, 2016. Characterization of the natural enemy community attacking cotton aphid in the Bt cotton ecosystem in northern China. Scientific Reports.)

7. We removed space between “aphelinids” and “and four species of aphelinids.”

8. We revised “ten different hyperparasitoid species were collected in cotton fields.” to “ten different hyperparasitoid species were collected from cotton fields.”

9. We keep the same trend for all scientific names.

10. For your question, “However, research on parasitoids in maize fields is still scarce. You might mean in northeastern? Or also one factor could be the change in transgenic maize introduction and now different species composition or dynamics or may be due to less area under cotton in northern China?”

Research on parasitoids in maize fields is still scarce in northeastern China.

11. We revised “ their parasitoids in 3 maize fields in 2018 and 2019 in northern China.” to “and their parasitoids in three maize fields during 2018 and 2019 seasons in northeastern China.”

12. “Could you add a map with these locations showing the northeastern locations?”

We had already added a map with these locations.Please see the revised manuscript.

13. “You may follow the paper published by Ali et al 2016/2018 for experimental design or sampling in details. You shall add the five sampling method reference as (Ali et al., 2018, 2016).”

We had already followed the paper published by Ali et al 2016/2018 for experimental design or sampling in details shall. We added detailed for experimental design and sampling in details. Please see the revised manuscript.

14. “Tables could be represented well in graphs rather tables and also tables could work for percentage.”

We have a lot of data, and we can clearly see the occurrence of aphids in each period with the table.

15. We have determined the consistency of objectives, methodology and results sections.

16. Is it Jilin province or norther eastern?

Our experimental fields were selected in the Jilin Province region of Northeast China

17. We had already gave future suggestions in the conclusion section.（Our next focus is to conduct laboratory studies on the parasitic function to clarify the reasons for the low parasitic rate recorded in the field and to continue to improve understanding of the parasitic bee species in the maize field.）

18. We revised “We are much thankful to our partners for their support of sample collection, insect rearing and laboratory assays” to “We are much thankful to our partners for their support in sample collection, insect rearing and laboratory assays.”

19. We checked the references carefully according to the format and spelling mistakes.

Sincerely 

Dr. Yuan

---

## [Decision Letter · Decision Letter 1]

3 Dec 2020

Population dynamics and species composition of maize field parasitoids attacking aphids in Northeastern China

PONE-D-20-32389R1

Dear Dr. yuan,

We’re pleased to inform you that your manuscript has been judged scientifically suitable for publication and will be formally accepted for publication once it meets all outstanding technical requirements.

Kind regards,

Yulin Gao

Academic Editor

PLOS ONE

---

## [Editor Report · Acceptance letter]

7 Dec 2020

PONE-D-20-32389R1 

Population dynamics and species composition of maize field parasitoids attacking aphids in Northeastern China 

Dear Dr. Yuan:

I'm pleased to inform you that your manuscript has been deemed suitable for publication in PLOS ONE. Congratulations! Your manuscript is now with our production department. 

Kind regards, 

on behalf of

Dr. Yulin Gao 

Academic Editor

PLOS ONE